# The slit diaphragm in *Drosophila* exhibits a bilayered, fishnet architecture

Deborah Moser [1,6], Konrad Lang [2,6], Alexandra N. Birtasu[1], Florian Grahammer[3,4], Martin Helmstädter[2,5], Margot P. Scheffer[1], Tobias Hermle [2,6] ✉ & Achilleas S. Frangakis [1,6] ✉

The kidney relies on the glomerulus to filter large volumes of blood plasma, with the slit diaphragm (SD) as a key structural component of the glomerular filtration barrier. Despite its central role, the molecular architecture of the SD has remained elusive for decades. Using cryo-electron tomography on focused ion beam-milled *Drosophila* nephrocytes, an invertebrate podocyte model, we show that the SD exhibits a bilayered fishnet architecture. In the cryo-electron tomography map, we observe criss-crossing strands spanning the extracellular space that can be populated with Sns and Kirre, the *Drosophila* orthologs of nephrin and Neph1, respectively. We show that *sns* silencing shortens the SD lines until disappearance, linking the fishnet architecture directly to Sns. After *Rab5* silencing, which causes Sns mistrafficking and ectopic formation of the SD, the fishnet pattern also appears ectopically. Elucidating the molecular SD architecture establishes a crucial link between the SD organization and its (patho)physiology.

The kidneys perform essential clearance of waste products from the blood, thereby generating about 180 liters of primary urine daily. The glomerulus acts as a molecular sieve that retains plasma proteins from blood filtered through three complementary layers: the fenestrated endothelium, the glomerular basement membrane and the slit diaphragm (SD), a specialized intercellular junction formed between the podocyte foot processes. The molecular framework of the SD is established by the immunoglobulin (Ig) domain-containing proteins nephrin[1–4] and Neph1[5,6], which interact extracellularly across the filtration slit[7,8]. These proteins assemble into a multi-protein complex featuring a hybrid composition of junctional components[9,10] that provides dynamic structural stability while integrating signaling cues[11,12]. DNA variants of *NPHS1*, the gene encoding nephrin, cause congenital nephrotic syndrome[1], highlighting the crucial role of the SD in proper kidney function.

Since the discovery of the SD in the mid-20th century, its molecular architecture has been the subject of debate[13]. Studying the SD is challenging, due to its absence from in vitro models, its limited accessibility, and the molecular complexity of its components. Using room temperature transmission electron microscopy (RT-TEM) on perfusion-fixed rodent kidneys, Rodewald and Karnovsky proposed a zipper-like structure for the SD[14]. Room temperature electron tomography showed molecular strands[15] and a layered, bipartite arrangement[16].

The podocyte-like nephrocyte in *Drosophila* is a genetically tractable model that enables mechanistic studies, variant analyses, and drug screening[17–20]. It has emerged as a valuable model of the glomerular filter due to its structural and molecular conservation[21–23]. As it features an accessible SD, *Drosophila melanogaster* is an ideal organism for investigating the molecular architecture of the SD.

Here we use cryo-focused ion beam (FIB)-milled plunge frozen nephrocytes to visualize the SD in situ. We reach a resolution sufficient to discern the molecular architecture formed by Sns and Kirre, the *Drosophila* orthologs of nephrin and Neph1, respectively. We show that

[1]Buchmann Institute for Molecular Life Sciences and Institute of Biophysics, Goethe University Frankfurt, Frankfurt, Germany. [2]Renal Division, Department of Medicine, Faculty of Medicine and Medical Center - University of Freiburg, Freiburg, Germany. [3]III. Department of Medicine, University Medical Center Hamburg-Eppendorf, Martinistraße, Hamburg, Germany. [4]Hamburg Center for Kidney Health (HCKH), University Medical Center Hamburg-Eppendorf, Hamburg, Germany. [5]EMcore, Medical Center - University of Freiburg, Freiburg, Germany. [6]These authors contributed equally: Deborah Moser, Konrad Lang, Tobias Hermle, Achilleas S. Frangakis. ✉e-mail: tobias.hermle@uniklinik-freiburg.de; achilleas.frangakis@biophysik.org

these proteins form a bilayered fishnet architecture. The fishnet pattern was abrogated by complete silencing of *sns*, the Drosophila nephrin. Acute *sns* silencing, reducing Sns to intermediate levels, produced shorter lines but the characteristic fishnet was maintained. This specific pattern further persisted in ectopic SDs formed upon acute *Rab5* silencing. Together, the genetic manipulations indicate that the fishnet architecture reflects the fundamental molecular arrangement of the SD proteins in a defined stoichiometry.

## Results

### Visualization of the *Drosophila* filtration barrier in situ

The *Drosophila* nephrocyte shapes membrane invaginations into an elaborate network of labyrinthine channels. Entry into these channels is regulated by a filtration barrier comprising a basement membrane and a SD[17,21,22]. The structural framework of the SD is established by

Sns and Kirre, analogous to the function of their human orthologs nephrin and Neph1[21,22] (Fig. 1a). The SD, forming the apical boundary along the furrow-like labyrinthine channels, creates a fingerprint-like pattern in tangential sections (Fig. 1b, Supplementary Fig. 1a, b), which corresponds to a regular, dot-like appearance in cross-sections (Fig. 1c, Supplementary Fig. 1c; 3D visualization in Fig. 1d, Supplementary Movie 1). The SD proteins Sns and Kirre colocalize at the entry points of the labyrinthine channels along the cell surface (Fig. 1e, g, Supplementary Fig. 1d–f). To gain further insight into the molecular architecture of the native *Drosophila* filtration barrier, we used cryo-electron tomography (cryo-ET) of cryo-FIB-milled, on-grid lamellae from freshly isolated nephrocytes (n = 6 garland nephrocyte chains). The samples were kept in their native frozen hydrated state throughout all the experiments. All characteristic landmarks of the *Drosophila* filtration barrier, including the basement membrane, the

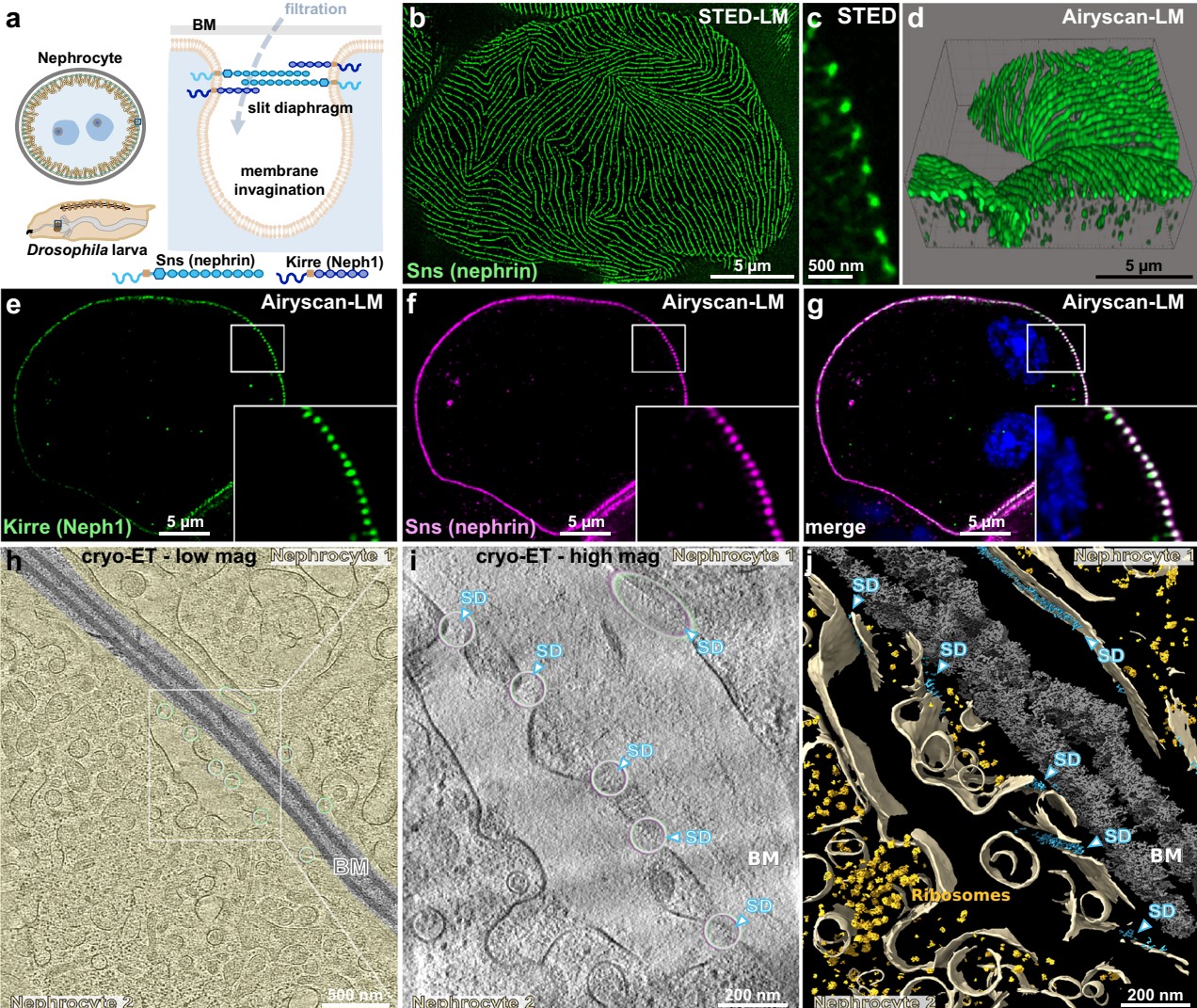

**Fig. 1 | Comparative analysis of the nephrocyte slit diaphragm (SD) using light microscopy and cryo-electron tomography. a** The schematic illustrates the anatomy and structure of nephrocytes including the SD, highlighting the two key SD proteins. (**b,c**) STED microscopy images of nephrocytes expressing Myc-tagged Sns from a CRISPR-edited gene locus[18]. Myc staining reveals a fingerprint-like pattern for the tagged SD protein in a tangential section (**b**) and a dot-like pattern in a cross-section (**c**). **d** Visualization using IMARIS software illustrates the relationship between linear and dot-like patterns based on Airyscan fluorescent microscopy of surface detail from nephrocytes stained for Myc-Sns. **e–g** Fluorescent microscopy of nephrocytes after co-labeling of (Myc)-Sns and Kirre shows colocalization of both SD proteins (enlarged inset). Nuclei are marked by Hoechst 33342 in blue. **h** A

3 nm thick computational slice of a low-magnification tomogram shows the cross-section of two nephrocytes. The cytoplasm of the cells is highlighted in light yellow, the basement membrane (BM) in gray. In the cytoplasm, several vesicles and cytoskeletal elements can be seen. Circles indicate regions containing SDs, where Sns and Kirre colocalize. **i** Computational slice of a tomographic reconstruction at higher magnification (1 nm pixel size) revealing the nephrocyte SD (highlighted by blue arrows and circles). The SD appears textured and not just as two lines connecting the plasma membranes. **j** Segmentation of the tomogram in (**i**), displaying the 3D architecture of the nephrocyte close to the cell surface, where SDs are found. Beige: membranes, blue: nephrocyte SDs, gray: basement membrane (BM), bright yellow: ribosomes.

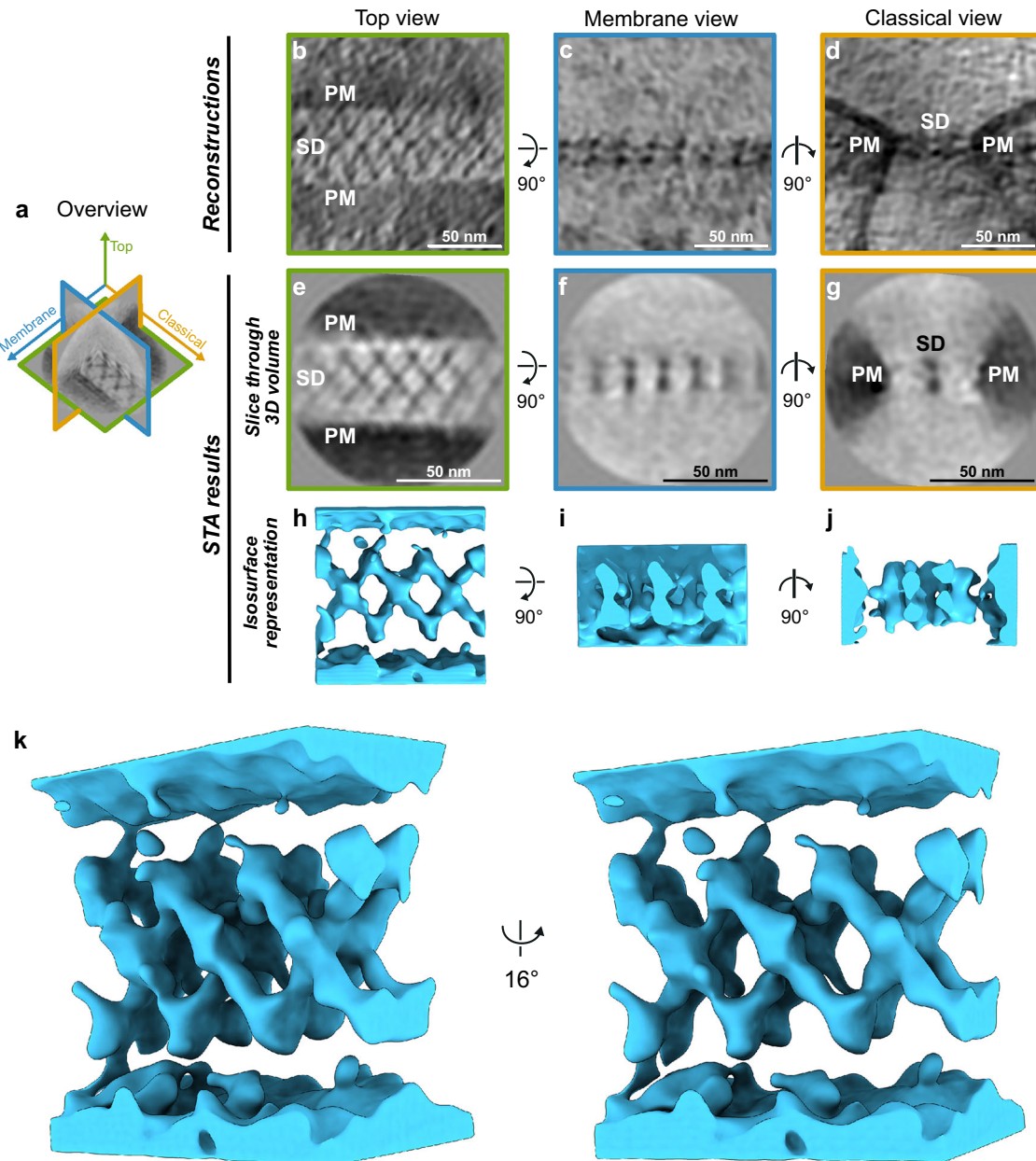

**Fig. 2 | Cryo-ET reveals that the slit diaphragm (SD) resembles a bilayered fishnet. a** Orthogonal planes through the SD average provide an overview of the 3D geometry. **b–d** 1 nm thick slices through the tomographic reconstructions show the nephrocyte SD from three different perspectives: **b** top view revealing the fishnet pattern of the SD between the two plasma membranes (PM); **c** membrane view revealing two parallel lines of "pearls on a string"; and **d** classical view displaying two parallel electron-dense lines reflecting the bilayered architecture of the SD. (**e–g**) 1 nm thick slices through the 3D volume of the SD obtained after sub-tomogram averaging, shown from the same perspectives as in (**b–d**) (pixel size: 2.7 Å). The slice showing the membrane view (**f**) illustrates the cross-section of the fishnet pattern at the center of the fishnet, parallel to the membrane. The slice showing the classical view (**g**) illustrates one of the cross-sections at the center of the fishnet perpendicular to the direction of the SD line, and thus appears as two dots. The result was obtained by averaging 595 particles selected from 16 tomograms, collected on 3 lamellae from three different *Drosophila* larvae and three different electron microscopy grids. **h–j** Isosurface representation of the cryo-ET map illustrating the 3D bilayered fishnet architecture, shown from the same perspectives as in (**e–g**) (pixel size: 2.7 Å). Individual strands criss-cross the space between the two plasma membranes, creating the fishnet pattern. **k** Stereo-pair of the SD at an oblique angle showing the bilayered architecture, with each of the layers resembling a fishnet.

labyrinthine channels that invaginate from the cell periphery, and the SDs at their tops, can be seen in three dimensions (3D) (Fig. 1h–j). The interior of each nephrocyte is filled with ribosomes, vesicles and various cytoskeletal elements (Fig. 1h–j).

## The SD in *Drosophila* exhibits a fishnet pattern
In the tomograms, the SD can be visualized in 3D from all possible viewpoints (Fig. 2, Supplementary Fig. 2, Supplementary Movie 2, 3 and 4). Three characteristic views reveal: (i) a fishnet pattern in which

individual ~52 nm long strands span the two plasma membranes at an angle of approximately 55 degrees (top view; Fig. 2b, Supplementary Fig. 2d–g); (ii) two parallel lines of "pearls on a string" densities (membrane view; Fig. 2c, Supplementary Fig. 2h–i); and (iii) two electron-dense parallel layers connecting the plasma membranes (classical view; Fig. 2d, Supplementary Fig. 2b, j), consistent with micrographs from RT-TEM[21]. The three views are orthogonal to each other and are shown in an isosurface visualization (Fig. 2h–k). The SD has a width of approximately 44 nm (Fig. 3a) in the extracellular space

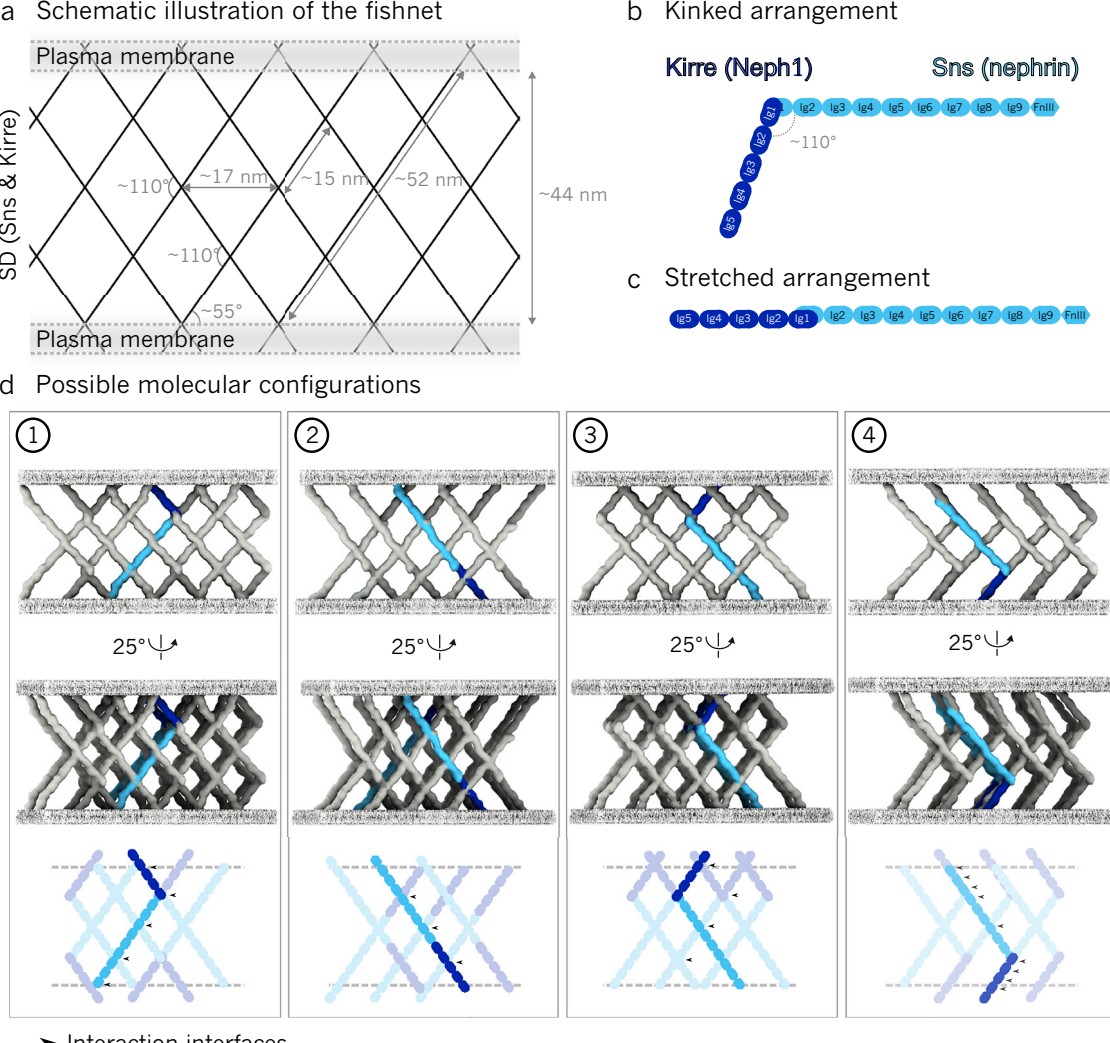

**Fig. 3 | Four possible configurations of Sns and Kirre within the slit diaphragm (SD). a** Blueprint of the densities observed by cryo-ET. In the middle of the SD, a prominent rhombus with an axis length of ~17 nm and an angle of ~110 degrees can be seen. All relevant dimensions and spacings are indicated. **b** Kinked arrangement of Sns and Kirre, as observed in the X-ray structure by Özkan et al[26]. **c** Stretched arrangement, where Sns and Kirre interact at an angle of 180°. **d** Four possible molecular configurations based on the fishnet pattern of the SD. Arrows indicate potential interaction interfaces between criss-crossing Ig domains.

between two adjacent plasma membranes, which is similar to what has been previously described using RT-TEM with chemical fixation[21,22]. Due to the significantly better resolution of cryo-ET compared to RT-TEM (Supplementary Fig. 3), it is possible to discern a periodic arrangement of the molecular densities composing the SD from the raw data alone (Fig. 1h-j, Fig. 2b-d, Supplementary Fig. 2, Supplementary Movie 2–4).

To enhance the signal from inherently noisy cryo-ET data, sub-tomogram averaging is commonly used. This approach involves averaging the signal of many identical objects, thereby improving the signal-to-noise ratio (SNR). By averaging 595 SD segments, we were able to discern a constant width of the labyrinthine channels of approximately 44 nm measured at the SD plane, and a periodic arrangement of the molecular strands at a higher contrast and resolution than in the raw data in the individual tomograms (Fig. 2e–j). The strands span from one plasma membrane to the other, creating a fishnet pattern. As views of the SD from several directions contributed to the average, the resulting cryo-ET map is isotropically resolved and depicts all SD views seen in the tomograms (Fig. 2). When visualized in 3D, the overall architecture resembles two fishnets, with one stacked on top of the other (Supplementary Movie 2). Each fishnet is created by criss-crossing strands. It can be sub-divided into triangular and

rhomboid shapes with side lengths of approximately 15 nm and angles of approximately 110 and 70 degrees (Fig. 2e–k, Fig. 3a).

## Four molecular configurations for Sns and Kirre are possible, but only one is structurally plausible

Next, we populated the cryo-ET map of the SD with Sns and Kirre. Sns and Kirre are both immunoglobulin (Ig) superfamily proteins, with Kirre presenting 5 Ig domains[24], and Sns 9 Ig domains and one FnIII-like domain[25]. Considering the dimensions of the SD (Fig. 3a) and the lengths of Kirre and Sns, one molecule alone is not long enough to span the distance between the two plasma membranes. Thus, Sns and Kirre proteins must interact in order to span the intermembrane distance. Since the resolution of the map is not sufficient to explicitly distinguish between Sns and Kirre, we constructed four possible molecular configurations for Sns and Kirre (Fig. 3d). Configuration 1 is in accordance with previous crystallographic studies and single-particle analysis studies in which the Ig1 domains of Sns and Kirre interact in a kinked arrangement with an angle of 90–110 degrees (Fig. 3b)[26]. This is consistent with the angle of approximately 110 degrees in the rhombus of the fishnet pattern (Fig. 3a). In configuration 2, Sns and Kirre interact in a stretched arrangement with an angle of 180 degrees (Fig. 3c). Although the interaction between the Ig1

domains of Sns and Kirre is predicted to be most stable at an angle of 90 to 110 degrees in solution, a different structural arrangement could be imposed in their native environment.

While configurations 3 and 4 are theoretically feasible, they appear biologically implausible. In configuration 3, Kirre and Sns would each derive from only one of the two membranes. In configuration 4, the thicknesses of individual strands are mismatched, creating clashes when two molecules are placed together (such as where multiple Ig domains overlap near the membrane). This configuration should result in a thicker density than we observe, and it should lack molecules in regions where we observe a density. Each proposed configuration generates specific potential interaction interfaces where individual Ig domains are close to each other and thus may interact (Table 1).

### Acute silencing of *sns* for 29 h reveals shorter SD lines, while the fishnet architecture of the SD is conserved

To examine the fishnet pattern forming under altered Sns/Kirre stoichiometry, we acutely silenced *sns* for 29 h, causing a reduction of the levels of the *Drosophila* nephrin to an intermediate level (Supplementary Fig. 4). Notably, the SD lines were shortened (Fig. 4a–c) but the fishnet architecture of the SD appeared undisturbed in the cryo-ET data (Fig. 4d–f). Direct comparison with the wild-type nephrocytes showed no difference, and no breaks in the overall periodicity were observed. This genetic background further allowed us to visualize start/end points of the fishnet-like pattern. The labyrinthine channels were seamlessly covered by the fishnet pattern at these points of initiation/termination. The unchanged pattern despite reduced Sns levels demonstrates that the fishnet architecture requires maintaining the stoichiometry between Sns and Kirre in the molecular structure.

### The fishnet pattern is abolished upon prolonged *sns* silencing

To examine whether the fishnet pattern requires the presence of the SD protein Sns, we examined animals showing full, rather than partial, loss-of-function of *sns*. Without GAL80-dependent temporal restriction, the continuous, and therefore more pronounced, silencing of *sns* abolished the characteristic Sns-derived signal in light microscopy (Supplementary Fig. 5). This confirms efficient silencing. When Kirre lacked its binding partner entirely, its linear arrangement was replaced by a pattern of clusters (Fig. 5a), while the mislocalized Kirre largely adhered to the surface (Fig. 5b). The altered configuration confirms that the Neph1 ortholog alone is insufficient for SD formation and consequently, the characteristic labyrinthine channels were no longer formed (Fig. 5c–e). In RT-TEM and cryo-ET acquisitions following such stronger, sustained silencing of *sns*, we observed electron-dense clusters decorating the cell membrane (Fig. 5c,e,f), but the SDs with their fishnet pattern were abrogated. We conclude that, consistent with previous findings[21,22], the presence of Sns is essential for the formation of the nephrocyte SD and is directly linked to the architecture of the fishnet pattern shown here.

### The fishnet SD shifts into the labyrinthine channels upon *Rab5*-RNAi-associated mistrafficking

Finally, we investigated a genetic background that exhibits mislocalized SDs without the loss of proteins directly associated with the SD. Knockdown of *Rab5*, a small GTPase involved in regulation of endocytosis and cargo sorting, is known to disrupt Sns trafficking and to lead to ectopic formation of the SD within the network of labyrinthine channels[18]. To avoid loss of cellular viability, we employed short-term *Rab5* silencing for 18 h and detected Sns in a mildly rarefied and blurred linear pattern in nephrocyte tangential sections (Fig. 6a). A cross-section also revealed Sns proteins protruding from the surface (Fig. 6b). This suggests the presence of ectopic SD proteins within the channel network, and SD formation was confirmed by RT-TEM (Fig. 6c). The bilayered fishnet pattern of the SD was maintained on the surface, as revealed by cryo-ET (Fig. 6d). With a spacing of approximately 17 nm between protein strands and a width of roughly 47 nm from membrane to membrane, the fishnet pattern corresponds to what we found for wild-type nephrocytes. Simultaneously, we observed the presence of SDs translocated from the cell surface deeper into the channel network (Fig. 6e, f), consistent with observations by light microscopy and RT-TEM (Fig. 6a–c). Thus, the disruption of Sns trafficking after *Rab5* silencing results in the ectopic appearance of the fishnet pattern. However, short-term silencing of *Rab5* seems to affect the localization of the SDs but not their structural assembly into a fishnet pattern.

## Discussion

The *Drosophila* nephrocyte SD exhibits a bilayered, fishnet architecture, which is absent following knockdown of *sns*, the *Drosophila* nephrin ortholog, and appears ectopically after SD mistrafficking. Partial silencing of *sns*, which disrupted the stoichiometry between the proteins Sns and Kirre, resulted in shorter lines, nevertheless exhibiting a fishnet architecture indistinguishable from wild type. This suggests that the fishnet architecture requires a precise ratio between the main structural components. The extent of the fishnet architecure followed availability of *sns*, breaking down entirely in its absence.

The architecture of the *Drosophila* SD is remarkably similar to that of the murine SD, which has been shown to also resemble a fishnet[27]. The fishnet architecture seems plausibly linked to the demanding functional requirements of the SD: Similar to modern day constructions like bridges, it can provide lateral and transversal mechanical stability between the plasma membranes while enabling a distinct spacing, thus allowing for size-selective permeability. A maximal mesh size of 17 nm would allow for the passage of albumin molecules, which directly corresponds to observations from tracer studies in *Drosophila*[23].

Recent studies indicate that the SD is a highly dynamic structure that undergoes rapid cycles of endocytosis and recycling[18]. The molecular arrangement of the SD constituents defines interaction interfaces between the Ig domains of Sns and Kirre. These interfaces may act as molecular guides, ensuring that newly delivered proteins are incorporated at their designated positions, while maintaining the structural integrity of the SD and facilitating its continuous turnover. This contrasts with earlier models, such as the zipper model[14], where removal of a single protein would compromise the stability by disrupting the lateral connections between SD constituents. Such a molecular backup as provided by a fishnet architecture could explain how continuous renewal of the SD occurs without protein leakage, despite constant filtration. At present, however, cryo-ET lacks temporal resolution, precluding direct observation of these dynamics at the spatial resolution necessary to visualize the fishnet pattern.

Previous studies involving staining with heavy metals could not achieve the resolution necessary to clearly discern individual strands in the SD. In cryo-ET, the sample is just frozen to liquid nitrogen temperatures without any subsequent treatment to enhance the contrast. With the technology we use here—cryo-ET of FIB-milled plunge-frozen samples—even atomic resolution has been achieved within cells[28]. In our study the resolution remains moderate due to: (i) The limited number of SD segments that we can record per tomogram, owing to the limited field of view and (ii) the inherent molecular flexibility of Sns und Kirre, the SD constituents, which do not allow for an even better averaging result, especially at the region close to the plasma membrane.

Different models have been proposed for the mammalian SD[14–16,27]. Single-particle electron microscopy and crystallography of SYG-1 and SYG-2, orthologs of nephrin (Sns) and Neph1 (Kirre), predicted an interaction angle of 90–110 degrees between the Ig1 domains of these proteins[26]. Notably, approximately 110 degrees is the angle that we measured between the criss-crossing strands

comprising the fishnet pattern. Given that both proteins are expressed from both sides of the channel and there are no molecular clashes, only configuration 1 appears plausible and consistent with the observed fishnet pattern (Fig. 3d). If Sns (nephrin) and Kirre (Neph1) are not straight but bent, several other configurations become possible. Each of the proposed configurations presents potential interaction interfaces between the Ig domains of Sns and Kirre that may facilitate the formation of the fishnet pattern. While it is known that Ig4, Ig6 and the FnIII-like domain of Sns are crucial for the formation of the SD[22], these characterized interactions do not unambiguously verify or exclude any of the four possible molecular configurations presented here. Therefore, gaining a deeper understanding of the potential interactions and affinities between the Ig domains is crucial for understanding the arrangement of proteins within the SD. Future mutation studies may also aid in determining which of the proposed configurations is correct.

Finally, our work highlights the relevance of *Drosophila* nephrocytes as a valuable podocyte model for studying SD biology and for addressing the gap left by the absence of this structure in in vitro models for mechanistic studies and drug discovery.

**Table 1 | Interaction interfaces between Sns and Kirre for each proposed configuration**

|  | Heterophilic interactions | Homophilic interactions |
|---|---|---|
| **Configuration 1** | Kirre-Ig1 and Sns-Ig1; Kirre-Ig4 and Sns-FnIII-like | Sns-Ig4 and Sns-Ig4; Sns-Ig7/8 and Sns-Ig7/8 |
| **Configuration 2** | Kirre-Ig1 and Sns-Ig1; Kirre-Ig2/3 and Sns-Ig6/7; Kirre-Ig5 and Sns-FnIII-like | Sns-Ig3 and Sns-Ig3 |
| **Configuration 3** | Kirre-Ig1 and Sns-Ig1 | Kirre-Ig4/5 and Kirre-Ig4/5; Sns-Ig4 and Sns-Ig4; Sns-Ig7/8 and Sns-Ig7/8 |
| **Configuration 4** | Kirre-Ig1 and Sns-Ig1; Kirre-Igs1-5 and Sns-Ig7-FnIII-like | Sns-Ig4 and Sns-Ig4 |

## Methods

### Fly strains and husbandry

Flies were raised on standard *Drosophila* food at 25 °C. *Prospero-GAL4*[21] was used to control expression of *sns*-RNAi (Bloomington Drosophila Stock Center #64872) in garland cell nephrocytes, and wild-type (yw[1118]) was crossed to *Prospero-GAL4* as control. For acute silencing of *Rab5* and *sns*, a fly stock carrying *dorothy*-GAL4 (BDSC #6903) and tubP-GAL80[ts] (BDSC #7019) was crossed with flies expressing either UAS-*Rab5*-RNAi (BDSC #34832) or UAS-*sns*-RNAi (Vienna Drosophila RNAi Center #109442[29]). Offspring larvae were initially raised at 18 °C to suppress RNAi expression, then shifted to 31 °C for 18 h (*Rab5*) or 29 h (*sns*) to induce acute RNAi expression. All stocks are listed in Supplementary Table 1.

### Cryo-electron tomography

**Sample preparation.** For cryo-electron tomography (cryo-ET), nephrocytes were dissected from *Drosophila* L3 larvae in Schneider's medium containing 10% glycerol[30]. During preparation, the tension applied to the nephrocytes themselves was kept at a minimum by cutting away the unwanted parts of the larva and carefully pipetting the nephrocytes onto the electron microscopy grids. The isolated nephrocytes were centrifuged (700 xg, 8 min) onto poly-L-lysin (PLL) coated EM grids (Plano, copper/palladium, G2019D 100 mesh, G2018 75 mesh, or G2050C 50 mesh). The EM grids were coated with Formvar to serve as a support film, and subsequently coated with PLL to facilitate adhesion of the nephrocytes. PLL coating was performed by glow-discharging the grids for 15 s, applying 8 μl of 0.1% (w/v) PLL in $H_2O$ (Sigma-Aldrich, St. Louis, Missouri, US), incubating for 20 min and washing them for 5 s in MilliQ® water[31–33]. After centrifugation, EM grids with nephrocytes were manually blotted (VWR, Qualitative filter paper, 413) and plunge frozen in liquid ethane (Vitrobot Mark IV, Thermo Fischer Scientific). The Vitrobot chamber was kept at 20 °C and 100% relative humidity. Samples were stored at −196 °C in liquid nitrogen.

**Cryogenic confocal laser scanning microscopy.** To facilitate sample targeting in the cryo-FIB-milling step, the plunge frozen grids were

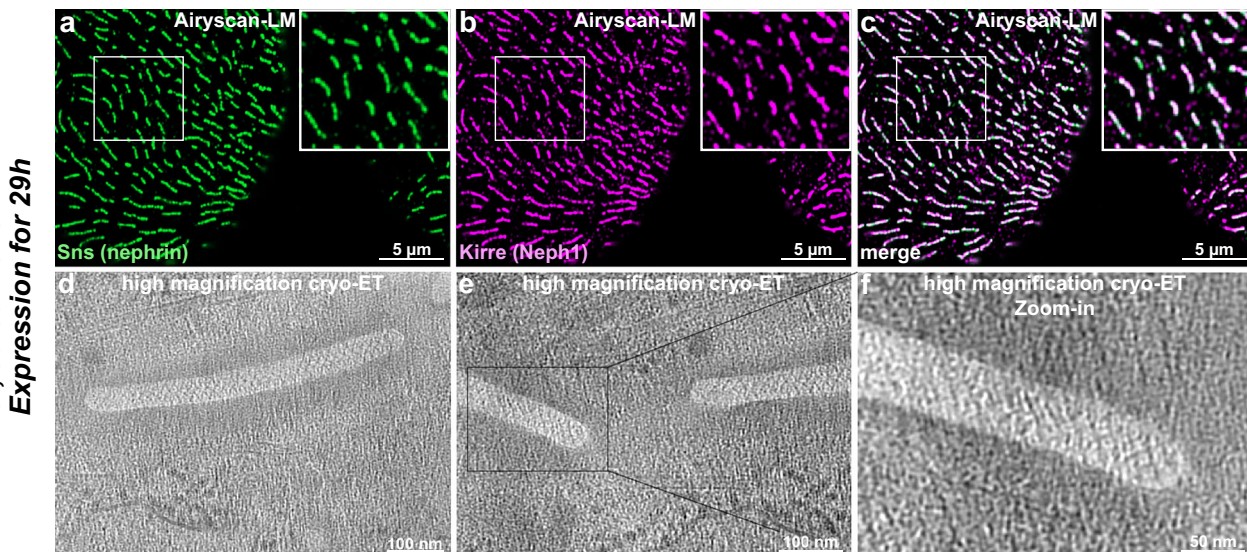

**Fig. 4 | Reduction of Sns to intermediate levels reveals shorter slit diaphragm (SD) lines, while the fishnet architecture of the SD remains. a–c** Fluorescence microscopy images acquired in Airyscan mode of nephrocytes after transient silencing of *sns* illustrate the reduction in the level of Sns (nephrin) protein, indicated by shortened lines of the SD. **d–f** Cryo-electron tomography confirmed the shorter SD lines, spanned by a fishnet pattern that is identical to the SD in wild-type nephrocytes. **d** 1 nm thick computational slice of a tomographic reconstruction illustrating a short SD line spanned by a fishnet-like SD. **e** Different region of interest of the same tomographic reconstruction as in (**d**), revealing the end or onset of two SD lines. **f** Zoom-in of the end/onset of the left line in (**e**). For *Dot;Gal80[ts]>sns*-RNAi nephrocytes, six tomograms displaying SDs could be acquired on 2 lamellae from two different *Drosophila* larvae on one electron microscopy grid.

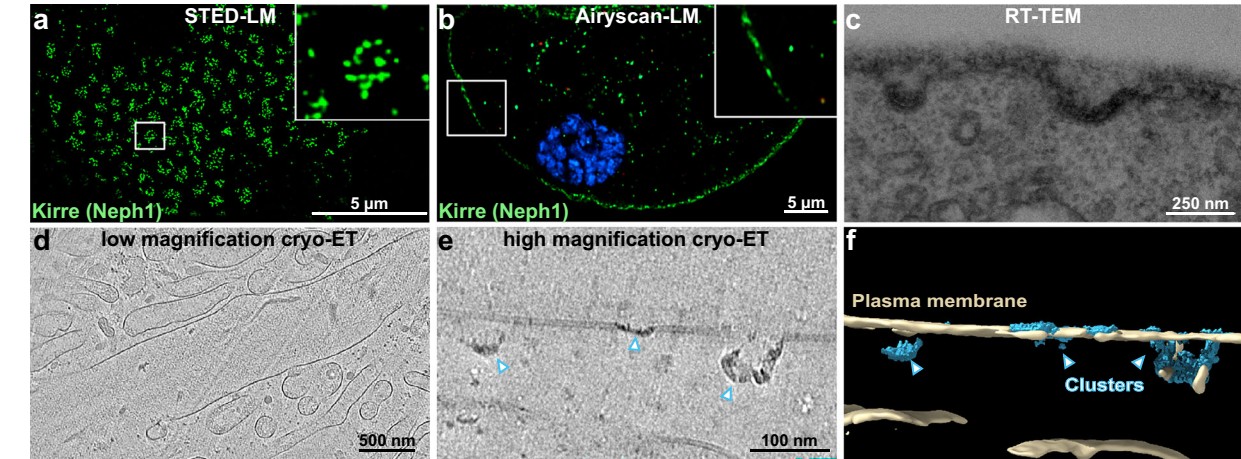

**Fig. 5 | In the absence of Sns, the fishnet architecture of the slit diaphragm (SD) is abolished. a,b** Fluorescence microscopy of nephrocytes expressing *sns*-RNAi persistently reveals a finely dotted pattern of Kirre in tangential sections (**a**) while the SD protein largely maintains an association with the cell surface, as indicated by cross-sections (**b**). **c** RT-TEM image shows detail of a nephrocyte expressing *sns*-RNAi. In the absence of Sns, SD formation is abrogated while electron-dense pits are formed on the surface. **d** A slice through the low-magnification tomographic reconstruction of a cross-section of two *sns* knockdown nephrocytes. No SD lines or labyrinthine channels can be identified on the cell surface. **e** A slice through the high-magnification tomographic reconstruction revealing electron-dense clusters close to the outer membrane of the nephrocyte. **f** Segmentation illustrates the 3D geometry of the clusters inside the plasma membrane of the nephrocyte (blue: clusters, beige: plasma membrane). For *pros>sns*-RNAi nephrocytes, 16 tomograms were acquired at the edge of the nephrocytes, where SDs would be expected to be formed. Tomograms were acquired on 4 lamellae of 3 *Drosophila* larvae on three different electron microscopy grids. Clusters could be found on two tomograms, no SD-like structures or channels could be seen on the other 14 tomograms.

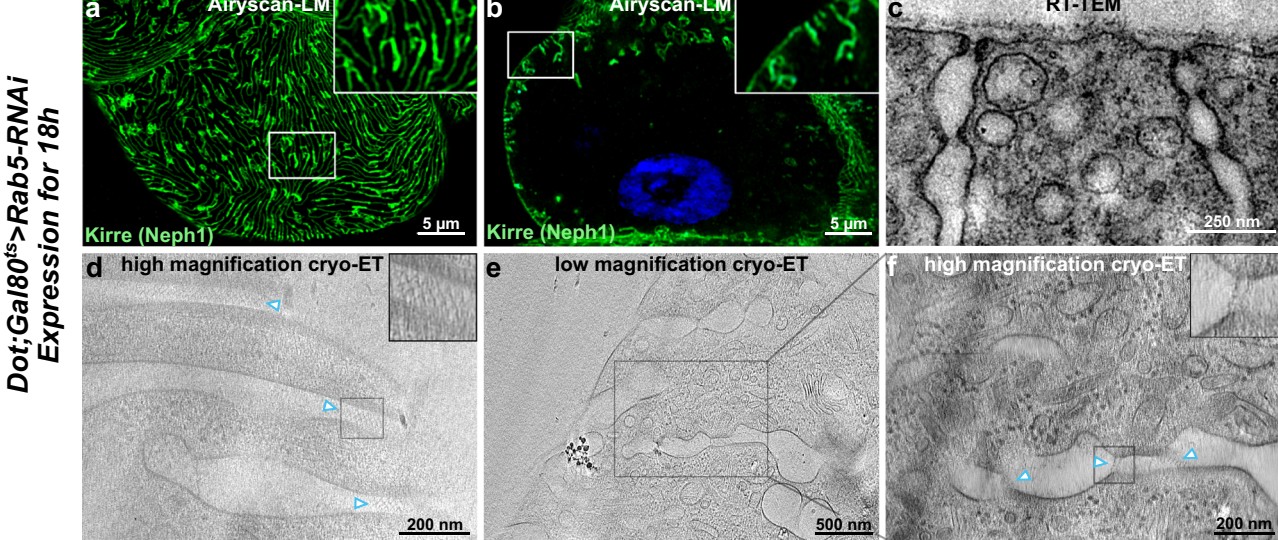

**Fig. 6 | The fishnet pattern appears ectopically upon *Rab5*-associated disruption of *sns* trafficking. a,b** Fluorescence microscopy of nephrocytes after short-term expression of *Rab5*-RNAi for 18 h shows a partially maintained SD pattern in a tangential section (**a**) and ectopic Sns protein protruding in lines from the nephrocyte surface in a cross-section (**b**). **c** An RT-TEM image of two labyrinthine channels of a nephrocyte expressing *Rab5*-RNAi for 18 h reveals ectopic SD formation. **d** A slice through the high-magnification tomographic reconstruction after cryo-ET of a nephrocyte expressing *Rab5*-RNAi shows the SD at the cell surface in top view. Insert: The zoom-in reveals the fishnet pattern, similar to that found in the wild-type nephrocyte. **e** A slice through the low-magnification tomographic reconstruction, where SDs can be found translocated towards the interior of the cell. **f** A slice of the high-magnification tomographic reconstruction of the same region of interest as in (**e**). Insert: The zoom-in reveals classical views of the SDs found in the nephrocyte expressing *Rab5*-RNAi. For *Rab5*-RNAi nephrocytes, seven tomograms displaying SDs could be acquired on 2 lamellae from two different *Drosophila* larvae on one electron microscopy grid.

imaged at −196 °C by confocal laser scanning microscopy (cryo-CLSM) (CMS196 cryo-stage, Linkam, Salfords, United Kingdom; LSM700, Carl Zeiss, Jena, Germany). Optical configurations were adjusted to capture the autofluorescence of the nephrocytes in the green channel, and the reflection from the grids in the far-red channel with excitation wavelengths of 488 and 639 nm, respectively. Images were acquired with a 5x/ NA 0.16 objective. Data acquisition was performed with Zeiss ZEN 2009 (blue) v2.1.

**Cryo FIB-milling.** After cryogenic confocal laser scanning microscopy, the plunge frozen grids were clipped into FIB autogrids (#1205101, Thermo Fisher Scientific Inc.). On-grid lamellae were produced by FIB-milling in a dual beam FIB-scanning electron microscope (SEM) (Helios Nanolab 600i, Thermo Fisher Scientific Inc.) following established protocols[27,34]. Correlation of fluorescence microscopy images with SEM images for targeting regions of interest was done using Fiji (v.2.9.0/1.53t)[35,36].

**Cryo-ET data acquisition**. Tilt series were acquired with SerialEM v4.1.0 beta[37] in a Titan Krios transmission electron microscope (Thermo Fisher Scientific Inc.) operated at 300 keV in nanoprobe EFTEM mode, equipped with a GIF Quantum S.E. post-column energy filter in zero-loss peak mode and a K3 detector (Gatan, Pleasanton, USA). Tilt series were recorded at a nominal magnification of 33,000x (1.34 Å per pixel) in super-resolution and dose-fractionation modes. A total of 41–52 images was acquired in dose-symmetric scheme using a tilt increment of 2° or 3°, starting from the pre-tilted lamella. Stage tilt angles ranged from −66° to +36°. The total dose per micrograph was at 2.5–3.125 e⁻/Å², and the defocus was set to −5 μm. Data acquisition parameters are summarized in Supplementary Table 2.

**Cryo-ET data processing**. Tilt series were processed and sub-tomogram averaging was performed using RELION-5[38]. Movies were motion corrected using the MotionCor2 wrapper[39] and CTF estimation was performed with CTFFIND-4.1.14[40]. Alignment was performed using the AreTomo2[41] wrapper or, alternatively, patch tracking in IMOD (v.4.11.24)[42]. Tomograms were reconstructed (binned pixel size 10.72 Å) and particles were manually selected and pre-oriented using ArtiaX (v.0.5) in UCSF ChimeraX (v.1.8)[43,44]. 848 sub-tomograms (box size 96, pixel size 10.72 Å) containing the two plasma membranes and the extracellular space with the SD were used for an initial 3D classification to select the best particles, yielding a list of 595 particles. Subsequent rounds of 3D refinement and gradual unbinning to a pixel size of 2.68 Å resulted in the final map at 39.58 Å. The mask used for refinement and postprocessing of the final cryo-ET map was generated using MATLAB R2022b[45]. Data processing parameters are summarized in Supplementary Table 3.

For Figs. 1h, 5d, e, 6e, and Supplementary Fig. 2a, the tomograms were reconstructed using IMOD's SIRT-like reconstruction (IMOD v.4.11.24)[42]. For Figs. 1i, 2b–d, Supplementary Fig. 2b, 3d, h, and Supplementary Movie 2, the tomograms were reconstructed using EmSART[46,47] and processed with tom_deconv[48] and IsoNet (v.0.2)[49]. For Fig. 4d, e tomograms were reconstructed in RELION-5[38] and denoised with the cryoCARE (v.0.1.1) wrapper[50] within RELION-5. For Fig. 6d, the tomogram was reconstructed in RELION-5[38]. For Fig. 6f, the tomogram was reconstructed in RELION-5[38], denoised with the cryoCARE (v.0.1.1) wrapper[50] within RELION-5, and subsequently processed with IsoNet (v.0.2)[49]. For Supplementary Fig. 2d–j and Supplementary Movies 3 and 4, tomograms were reconstructed using EmSART[46,47] and processed with tom_deconv[48]. Segmentations were generated with Dragonfly software, Version 2024.1, for Windows[51], and edited with mcm-cryoet[52].

**Generation of 3D Models for the arrangement of Sns and Kirre within the SD**. The molecular structure of Sns (UniProtKB A1Z7J1) and Kirre (UniProtKB Q9N9Y9) was predicted using AlphaFold3[53]. The intracellular, transmembrane and signal peptide sections of the Sns and Kirre AlphaFold predictions were removed, and Sns (residues 71-1069) and Kirre (residues 81-568) were then adapted to the cryo-ET density using Coot (v.0.9.8)[54]. Subsequently, for the "kinked arrangement" (Fig. 3b) Sns and Kirre were assembled into a heterodimer based on the crystal structure of the SYG-1-SYG-2 heterodimer (PDB ID: 4OFY)[26] using UCSF ChimeraX (v.1.8)[44]. For the "stretched arrangement" (Fig. 3c), Kirre was rotated by ~70° keeping the interaction interface between Sns-Ig1 and Kirre-Ig1 as similar as possible as in the "kinked arrangement". The aligned heterodimers were fitted into the cryo-ET density map to generate our 4 proposed models (Fig. 3d) using ChimeraX. An idealized density map at 20 Å resolution was generated for each configuration using the ChimeraX molmap command and colored according to our schematics for Sns and Kirre. Lipid patches were generated using the CHARMM-GUI[55].

## Conventional transmission electron microscopy

For conventional transmission electron microscopy (TEM), nephrocytes were dissected in phosphate buffered saline (PBS) followed by fixation in a mix of 4% paraformaldehyde, 2% glutaraldehyde, and 0.1 M cacodylate buffer pH 7.4. TEM was carried out using standard techniques.

## Immunofluorescence studies

For immunofluorescence, the nephrocytes were dissected and fixed in 4% paraformaldehyde in PBS for 20 min. Subsequent antibody labeling was performed according to standard procedures. All primary antibodies used in this study are listed in Supplementary Table 4. Nuclei are marked by Hoechst 33342 (Sigma, B2261; 1:500-1:1000). Samples were imaged using a Zeiss LSM 980 applying Airyscan mode or an Abberior STEDYCON for stimulated emission depletion (STED) microscopy. For light microscopy images, 6–8 animals per genotype were imaged per experiment, with experiments repeated at least three times. For each genotype, a representative image was selected to be displayed in the figure.

## Reporting summary

Further information on research design is available in the Nature Portfolio Reporting Summary linked to this article.

## Data availability

No large-scale data sets from high-throughput analyses were generated or analyzed in this study. The cryo-ET structure solved in this study is available in the Electron Microscopy Data Bank (EMDB) under the accession code EMD-53557. Atomic coordinates of the previously determined X-ray structure used in this study is available in the Protein Data Bank (PDB) under the following accession code: 4OFY (SYG-1 and SYG-2 complex). Unprocessed confocal or electron microscopy images are available upon request. All remaining data are included within the manuscript.

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

## Acknowledgements

We are grateful to Gerd Walz for valuable advice. We thank Charlotte Meyer and Séverine Kayser for outstanding technical support. We are

grateful to F. Ditengou, Lighthouse Core Facility, University of Freiburg, for support regarding confocal microscopy. Lighthouse Core Facility is funded in part by the Medical Faculty, University of Freiburg (Project Numbers 2021/A2-Fol; 2021/B3-Fol) and the Deutsche Forschungsgemeinschaft (DFG; Project Number 450392965). We thank the Frankfurt Center for Electron Microscopy for measurement time. We are grateful to Lilli Skaer for assistance with cryo-electron tomography sample preparation. We thank Pauline Roth for technical support and software development to facilitate processing of cryo-electron tomography data. We thank Maria Ericsson (Harvard Medical School Electron Microscopy Facility) for technical assistance. We thank the Bloomington Drosophila Stock Center and the Vienna Drosophila RNAi Center for providing fly stocks. The Core Facility for Electron Microscopy (EMcore) at the University Freiburg Medical Center—IMITATE is registered with the DFG under the reference number RI_00555. DFG provided funding for F.G. (CRC 1192, GR3933/1-1, GR3933/1-2, TRR-422), M.P.S. (FR 1653/14-1), T. H. (HE 7456/7-1 and project-ID 431984000 – SFB 1453). T.H. acknowledges support from the Heisenberg Program of the DFG (HE 7456/6-1). The DFG funded the Research Training Group iMOL (GRK 2566/1) for D. M. and A.N.B.

## Author contributions

D.M., K.L., A.N.B. and M.H.: prepared samples for cryo-electron tomography (cryo-ET). D.M.: prepared the FIB-milled lamellae, acquired cryo-ET data and processed the data. K.L.: reared the flies, prepared samples for light microscopy, and acquired confocal and stimulated emission depletion microscopy data. D.M., K.L., A.N.B. and T.H.: prepared the data figures. M.P.S.: designed the research, performed processing and refined the final density map. T.H. and M.H.: prepared samples for room temperature transmission electron microscopy (RT-TEM) and acquired RT-TEM data. A.S.F., F.G. and T.H.: designed and supervised the research. D.M., K.L., T.H. and A.S.F.: wrote the manuscript, with contributions from all authors.

## Funding

## Competing interests

The authors declare no competing interests.
