## [Transparent Peer Review file · Nature Communications]

The slit diaphragm in *Drosophila* exhibits a bilayered, fishnet architecture

Corresponding Author: Professor Achilleas Frangakis

Version 1:

Reviewer comments:

Reviewer #1

(Remarks to the Author)

Review of "The slit diaphragm in *Drosophila* exhibits a bilayered, fishnet architecture" by Deborah Moser, Konrad Lang, Alexandra N. Birtasu, Florian Grahammer, Margot P. Scheffer, Martin Helmstädter, Tobias Hermle, and Achilleas S. Frangakis

The authors present the structure of the slit diaphragm from *Drosophila* podocytes. The structure was observed with cryo electron tomography of FIB-milled, plunge frozen, isolated nephrocytes. They also propose a well-presented model for how the known slit diaphragm proteins *sns* and *kirre* (*Drosophila* orthologs of human Neph1 and Neph1) fit into this density. They perform silencing of *sns* and *Rab5*. The former causes the fishnet structure to be abolished. The latter is a trafficking perturbation that causes the fishnet pattern to be established ectopically in intracellular labyrinthine channels within the podocyte. This is an important model system and it is necessary to thoroughly understand the structure not only to better understand the mammalian slit diaphragm structure but also so that future work on the model system can be put in context. The view of this reviewer is that the reporting of this structure is straight-forward, well-done, and should be considered an important step worthy of publication. While the presented structure is mostly descriptive and definitely could be more deeply and quantitatively explored, the time-consuming nature of the FIB-milled cryoET experiment makes it reasonable to expect this from future manuscripts. Specific concerns are below.

- 1) In multiple places (including the abstract), the authors compare this work to their own reported structure of the murine slit diaphragm that was posted on BioRxiv in 2023. It is problematic to reference this work so heavily given that it hasn't been peer reviewed. As the work is two years old, the authors should give an explanation as to where this work is in the publication process. If it is not going to be peer-reviewed, I do not believe it is appropriate to use it as evidence or comparison for this manuscript.
- 2) Given that the sub-tomogram averaging is low-resolution and many mesh-like artifacts are possible with averaging techniques, it would be very pertinent to supply the reader with as many raw examples from tomogram reconstructions as possible. The one WT tomogram in the manuscript is also the movie example. It is striking and represents the presented structure well. Please show many other examples.
- 3) The authors mention the thickness of the fishnet strands in multiple places. This is quite difficult to determine given the resolution of the structure being reported. It should be softened.
- 4) This structure looks like it could change considerably with stretching. Please try to describe heterogeneity or homogeneity of the structure being presented. Also, please make an effort to discuss the possibility of these forces being modified due to plunge freezing, glycerol, and cell isolation.
- 5) A mutation that modifies or adds or removes density from the structure of either *sns* or *kirre* would have been much more informative (given the story being told in the first half of the manuscript) than the silencing. Please strengthen your explanation for why the silencing experiments were necessary to understand the structure or explain the pivot.
- 6) The enclosed checklist says that statistical N's are given in the main body of the manuscript but I was not easily able to find them. The number of grids, lamellae and tomograms are very helpful numbers for interpreting the manuscript. The numbers should be in the figure legends.
- 7) (minor) The paragraph about cryoET in the discussion seems a bit out of place. Stating that it is possible to get atomic-level detail from sub-tomogram averaging seems misleading in this particular situation given that you aren't following it up with an explanation of the distinct limitations of this particular case (sample size and structural flexibility).

Congratulations on some beautiful imaging!

Reviewer #2

(Remarks to the Author)

Deborah Moser et al. used cryo-electron tomography to investigate slit diaphragm (SD) architecture of fly nephrocytes, the equivalent of kidney podocytes. They found that the slit diaphragms of *Drosophila* nephrocyte form a fishnet structure, which was not found in previous studies. The authors showed that the fishnet is bilayered, which is consistent with the TEM data by Weavers et al. 2009 (PMID: 18971929). Remarkably, the fishnet pattern shows high similarity with mouse glomerulus SD reported by Birtasu et al. 2023 (<https://doi.org/10.1101/2023.10.27.564405>), indicating that the filtration structure is highly conserved from fly to mammals. The reveal of the fishnet structure largely advanced the understanding of the filtration structure and function.

The images are in high resolution and good quality; however, Birtasu's cryo-EM data showed the fishnet structure of mouse SD (bioRxiv 2023/JASN2024), reducing the novelty of this study. They described the fishnet structure and the potential molecular components; the underlying molecular structure is largely based on prediction.

Major comments:

1. Cryo-EM was used in both Birtasu's work and this study. Birtasu et al. found a fishnet pattern in mouse glomerulus, in this study Moser et al. found a similar fishnet structure in fly nephrocyte. The authors mentioned in the discussion that the bilayered fishnet "may act as a template to guide newly delivered proteins during constant turnover". The authors can take advantage of fly genetics to address this, as such adding in more novelty.
2. Lang et al. 2022 (PMID: 35876643) used a temperature sensitive TARGET system (Dot-Gal4, Gal80ts) to manipulate sns expression and nicely showed that (Dot-Gal4, Gal80ts>sns-RNAi) caused reduced SD. In this manuscript Figure 4 G-L, the authors used (Dot-Gal4, Gal80ts) to manipulate Rab5 expression. However, in the same figure, they used pros driver to silence sns (panels A-F) and observed loss of fishnet pattern. Why different drivers were used to downregulate sns and Rab5, respectively? Previous studies (e.g., PMID: 18971929) showed that Sns depletion caused loss of slit diaphragm. Thus, it is expected that the fishnet is absent in pros>sns-RNAi nephrocytes. Will it provide more information if the authors use (Dot-Gal4, Gal80ts) to silence sns at developmental stages or adult stages?
3. The images in Figure 4 panel J and supplementary Figure 2 panel B show high similarities. A clarification is missing in supplementary Figure 2 legend to show the genotypes.
4. It is challenging to appreciate the structure in figure 3D. For a comparison, the fishnet pattern proposed by Birtasu et al. 2023 is easy to understand.

Minor comments:

1. In figure 1, panel F shows a region where two nephrocytes form contacts. This is different from the enlarged insets in panels E-E". What is the rationale to choose the distinct regions?
2. In figure 1 F and G, the image contrast does not seem to be optimal. Is there a way to improve it so that the reader can better appreciate the fine structures?

Version 2:

Reviewer comments:

Reviewer #1

(Remarks to the Author)

Second review of "The slit diaphragm in *Drosophila* exhibits a bilayered, fishnet architecture" by Deborah Moser, Konrad Lang, Alexandra N. Birtasu, Florian Grahammer, Margot P. Scheffer, Martin Helmstädter, Tobias Hermle, and Achilleas S. Frangakis

All of my previous concerns were addressed. This manuscript is improved and well-suited for publication. A few observations are:

- 1) The light yellow coloring in Fig. 1H is not really visible.
- 2) "Persistently" is spelled incorrectly on line 228.
- 3) The transition between the section on partial silencing and more complete silencing of sns is confusing because it sounds like you are presenting something totally different rather than just doing the silencing longer. I think it needs to be just clearly stated in the first sentence that the previous silencing system was prolonged to get a more pronounced effect.
- 4) The way you discuss the ectopic formation of SD in labyrinthine channels is confusing to me. In the non-silenced sample, you described the correctly localized SD as being in labyrinthine channels so it is confusing to say that it then gets ectopically localized to labyrinthine channels.

It was a pleasure reviewing your manuscript.

Reviewer #2

(Remarks to the Author)

In the revised version, Deborah Moser et al. provided additional data to show the fishnet structure of the nephrocyte slit

diaphragm. Interestingly, reducing Sns to intermediate levels results in shorter slit diaphragm (SD) lines, while the overall fishnet architecture of the SD is preserved. The new images are also of higher quality. Overall, the authors have addressed the concerns appropriately.

We would like to thank both reviewers for their immensely valuable feedback, which helped us improve the manuscript. We are committed to upload our micrographs and tilt series to the EMPIAR database (<https://www.ebi.ac.uk/empiar/>), so that the raw data will be freely available.

The reviewer comments are in black, and our responses are in navy blue.

REVIEWER COMMENTS

Reviewer #1 (Remarks to the Author):

Review of “The slit diaphragm in *Drosophila* exhibits a bilayered, fishnet architecture” by Deborah Moser, Konrad Lang, Alexandra N. Birtasu, Florian Grahammer, Margot P. Scheffer, Martin Helmstädter, Tobias Hermle, and Achilleas S. Frangakis

The authors present the structure of the slit diaphragm from *Drosophila* podocytes. The structure was observed with cryo electron tomography of FIB-milled, plunge frozen, isolated nephrocytes. They also propose a well-presented model for how the known slit diaphragm proteins *sns* and *kirre* (*Drosophila* orthologs of human Neph1 and Neph1) fit into this density. They perform silencing of *sns* and *Rab5*. The former causes the fishnet structure to be abolished. The latter is a trafficking perturbation that causes the fishnet pattern to be established ectopically in intracellular labyrinthine channels within the podocyte. This is an important model system and it is necessary to thoroughly understand the structure not only to better understand the mammalian slit diaphragm structure but also so that future work on the model system can be put in context. The view of this reviewer is that the reporting of this structure is straight-forward, well-done, and should be considered an important step worthy of publication. While the presented structure is mostly descriptive and definitely could be more deeply and quantitatively explored, the time-consuming nature of the FIB-milled cryoET experiment makes it reasonable to expect this from future manuscripts. Specific concerns are below.

1) In multiple places (including the abstract), the authors compare this work to their own reported structure of the murine slit diaphragm that was posted on BioRxiv in 2023. It is problematic to reference this work so heavily given that it hasn't been peer reviewed. As the work is two years old, the authors should give an explanation as to where this work is in the publication process. If it is not going to be peer-reviewed, I do not believe it is appropriate to use it as evidence or comparison for this manuscript.

We thank the reviewer for this comment and we agree with their perspective. Indeed, this work is in the publication process, which is taking exceptionally long. Birtasu et al. will certainly be published.

We refer now to this work only once in the discussion, as indeed the fishnet pattern (although more rudimentary than here) was first published online as a preprint:

“The architecture of the *Drosophila* SD is remarkably similar to that of the murine SD, which has been shown to also resemble a fishnet²⁷.”

In addition to a later comment, the fact that both systems display a similar architecture (fishnet), despite having a different structure, shows the importance of this structure.

2) Given that the sub-tomogram averaging is low-resolution and many mesh-like artifacts are

possible with averaging techniques, it would be very pertinent to supply the reader with as many raw examples from tomogram reconstructions as possible. The one WT tomogram in the manuscript is also the movie example. It is striking and represents the presented structure well. Please show many other examples.

We thank the reviewer for this suggestion, which we implemented one-to-one. Indeed, we overlaid the raw data and the averaged density, and they look comparable. We have now added **Supplementary Figure 2**, where we show examples of the slit diaphragm in eight different tomograms from three different flies, as well as **Supplementary Movie 3** and **4**, illustrating the three dimensional volume of the tomograms shown in **Supplementary Figure 2a-c**, and **2f**, respectively.

Additionally, we are committed to upload our micrographs and tilt series to EMPIAR, so that the raw data will be freely available to the scientific community.

3) The authors mention the thickness of the fishnet strands in multiple places. This is quite difficult to determine given the resolution of the structure being reported. It should be softened.

We thank the reviewer for pointing this out. We have now softened the statement and contextualized it with the resolution at all the instances in our manuscript where this was the case.

4) This structure looks like it could change considerably with stretching. Please try to describe heterogeneity or homogeneity of the structure being presented.

Indeed, we measured the amount of stretching as the distance between the two labyrinthine channel membranes which appears surprisingly constant. Although an automated classification of the variability of the extracted subtomograms is not possible due to the limited sample size, it is probable that the resolution of the structure presented here is limited due to small changes in the fishnet-like architecture. We observe some degree of variation in the shape and size of the rhomboids in the fishnet pattern in the raw data (within the reported resolution). From the constant membrane distance we conclude that the particles extracted from the data are overall homogeneous within the reported resolution. We now also address this in the results:

“By averaging 595 SD segments, we were able to discern a constant width of the labyrinthine channels of approximately 44 nm measured at the SD plane and a periodic arrangement of the molecular strands at a higher contrast and resolution than in the raw data in the individual tomograms.”

Also, please make an effort to discuss the possibility of these forces being modified due to plunge freezing, glycerol, and cell isolation.

We thank the reviewer for this comment. Cryo-EM sample preparation is known to provide optimal preservation of native structural features. The methods that we used for sample preparation have been previously established in the field. Usage of glycerol for cryo-preservation prior plunge freezing has for instance been shown for on-grid cell cultures (Franken et al., 2022, PMID: 36149798). Usage of glycerol in combination with high pressure

freezing has also been shown for brain tissue samples, where cellular and subcellular features were shown to be preserved (Creekmore et al., 2024, PMID: 38531877). A similar protocol to the one used in this manuscript has also been shown for *Drosophila* tissue (Bäuerlein et al., 2023, <https://doi.org/10.1101/2021.04.14.437159>), portraying well conserved cellular structures such as mitochondria and ER.

Regarding cell isolation, we now also mention in the Methods: “During preparation, the tension applied to the nephrocytes themselves was kept at a minimum by cutting away the unwanted parts of the larva and carefully pipetting the nephrocytes onto the electron microscopy grids.”

5) A mutation that modifies or adds or removes density from the structure of either *sns* or *kirre* would have been much more informative (given the story being told in the first half of the manuscript) than the silencing.

Indeed this would be the ideal way of addressing this. However, clinical mutations causing malformed slit diaphragms have not been reported and it is difficult to predict variants that induce a structural change without disrupting the formation of the SD entirely.

To address the reviewers’ request, we opted for acute silencing of *sns* to reduce its protein levels to an intermediate range, enabling us to study a partial loss-of-function.

We have now included in the manuscript a complete subsection: “Acute silencing of *sns* for 29h reveals shorter SD lines, while the fishnet architecture of the SD is conserved”.

Please strengthen your explanation for why the silencing experiments were necessary to understand the structure or explain the pivot.

The goal of the silencing experiments was to directly correlate the fishnet pattern with the presence of *Sns* and the slit diaphragm, while ruling out an artificial origin of the pattern. Therefore, we first examined animals with *sns* knockdown. In this background lacking the *Drosophila* *nephrin*, the fishnet pattern was completely absent, demonstrating that the pattern is specifically dependent on the presence of *Sns*. This is further supported by *Rab5*-RNAi, which is known to cause formation of slit diaphragms deeper within the channels. Accordingly, the fishnet pattern appeared ectopically within the labyrinthine channels. Taken together, this clearly demonstrates a direct association of the pattern with the slit diaphragm, as well as ruling out an artifact. We have rephrased our conclusion on the *sns* silencing experiment to convey the link between the presence of *Sns* and the fishnet architecture more clearly:

“We conclude that, consistent with previous findings^{21,22}, the presence of *Sns* is essential for the formation of the nephrocyte SD and is directly linked to the architecture of the newly discovered fishnet pattern.”

6) The enclosed checklist says that statistical N’s are given in the main body of the manuscript but I was not easily able to find them. The number of grids, lamellae and tomograms are very helpful numbers for interpreting the manuscript. The numbers should be in the figure legends.

We have appended all relevant numbers both in the manuscript and in the figure legends as the reviewer suggested.

For the wild-type nephrocytes we have now included the number of particles used for averaging, the total number of tomograms, lamellae, flies, and electron microscopy grids that were used to produce the final structure shown in the figure legend for **Figure 2**. Similarly, in the figure legend for **Figure 4, 5, and 6** we have also included the number of EM Grids, flies, lamellae, and tomograms for each of the genotypes.

7) (minor) The paragraph about cryoET in the discussion seems a bit out of place. Stating that it is possible to get atomic-level detail from sub-tomogram averaging seems misleading in this particular situation given that you aren't following it up with an explanation of the distinct limitations of this particular case (sample size and structural flexibility).

We have extended this paragraph to address the limitations of this study, since the structural flexibility is directly related to the achievable resolution:

“Previous studies involving staining with heavy metals could not achieve the resolution necessary to clearly discern individual strands in the SD. In cryo-ET, the sample is just frozen to liquid nitrogen temperatures without any subsequent treatment to enhance the contrast. With the technology we use here – cryo-electron tomography of FIB-milled plunge-frozen samples – even atomic resolution has been achieved within cells²⁸. In our study the resolution remains moderate due to: (i) The limited number of SD segments that we can record per tomogram, owing to the limited field of view and (ii) the inherent molecular flexibility of Sns und Kirre, the SD constituents, which do not allow for an even better averaging result, especially at the region close to the plasma membrane.”

Congratulations on some beautiful imaging!

We thank the reviewer for the extremely constructive comments that helped us improve the manuscript.

Reviewer #2 (Remarks to the Author):

Deborah Moser et al. used cryo-electron tomography to investigate slit diaphragm (SD) architecture of fly nephrocytes, the equivalent of kidney podocytes. They found that the slit diaphragms of *Drosophila* nephrocyte form a fishnet structure, which was not found in previous studies. The authors showed that the fishnet is bilayered, which is consistent with the TEM data by Weavers et al. 2009 (PMID: 18971929). Remarkably, the fishnet pattern shows high similarity with mouse glomerulus SD reported by Birtasu et al. 2023 (<https://doi.org/10.1101/2023.10.27.564405>), indicating that the filtration structure is highly conserved from fly to mammals. The reveal of the fishnet structure largely advanced the understanding of the filtration structure and function.

The images are in high resolution and good quality; however, Birtasu's cryo-EM data showed the fishnet structure of mouse SD (bioRxiv 2023/JASN2024), reducing the novelty of this study. They described the fishnet structure and the potential molecular components; the underlying molecular structure is largely based on prediction.

Major comments:

1. Cryo-EM was used in both Birtasu's work and this study. Birtasu et al. found a fishnet pattern in mouse glomerulus, in this study Moser et al. found a similar fishnet structure in fly nephrocyte. The authors mentioned in the discussion that the bilayered fishnet “may act as a

template to guide newly delivered proteins during constant turnover”. The authors can take advantage of fly genetics to address this, as such adding in more novelty.

At this point it is not possible to observe the highly dynamic turnover at the resolution required to resolve the net-like molecular structure. Originally our statement was intended to distinguish the fishnet model resulting from our data from previous models of the SD, which provide limited insight into the turnover mechanism.

In the “zipper model” (Rodewald and Karnovsky, 1974, PMID: 4204974), removal of one molecule would cause the whole SD to “unzip”, as the lateral connections between the SD constituents would immediately be lost. Additionally, newly translocated molecules would need to locate its binding partners from both sides, a process that, at least from a structural biological perspective, seems challenging if not unlikely.

In the “spring model” (Grahammer et al. 2016, PMID: 27430022), it still remains unclear how Sns and Kirre may interact with the opposing membrane. Further, Sns and Kirre, both highly flexible molecules, would need to stretch and extend to their complete length to reach the opposing membrane, and interact with a - despite elaborate proteomics studies - yet unknown partner molecule or interaction site.

In response to the reviewer’s comments, we have expanded the discussion to address the limitation imposed by the lack of temporal resolution in cryo-ET, and to outline how our fishnet architecture may provide a structural framework to better understand SD turnover:

“Recent studies indicate that the SD is a highly dynamic structure that undergoes rapid cycles of endocytosis and recycling¹⁸. The molecular arrangement of the SD constituents defines interaction interfaces between the Ig domains of Sns and Kirre. These interfaces may act as molecular guides, ensuring that newly delivered proteins are incorporated at their designated positions, while maintaining the structural integrity of the SD and facilitating its continuous turnover. This contrasts with earlier models, such as the zipper model¹⁴, where removal of a single protein would compromise the stability by disrupting the lateral connections between SD constituents. Such a molecular backup as provided by a fishnet architecture could explain how continuous renewal of the SD occurs without protein leakage, despite constant filtration. At present, however, cryo-ET lacks temporal resolution, precluding direct observation of these dynamics at the spatial resolution necessary to visualize the fishnet pattern.”

2. Lang et al. 2022 (PMID: 35876643) used a temperature sensitive TARGET system (Dot-Gal4, Gal80ts) to manipulate sns expression and nicely showed that (Dot-Gal4, Gal80ts>sns-RNAi) caused reduced SD. In this manuscript Figure 4 G-L, the authors used (Dot-Gal4, Gal80ts) to manipulate Rab5 expression. However, in the same figure, they used pros driver to silence sns (panels A-F) and observed loss of fishnet pattern. Why different drivers were used to downregulate sns and Rab5, respectively?

The choice of driver lines was guided by the specific experimental goals. For *Rab5* silencing (Figure 6), we aimed to induce an acute loss-of-function to observe slit diaphragm mistrafficking. Stronger and sustained RNAi expression targeting *Rab5* leads to a severely compromised cellular state lacking slit diaphragms entirely (PMID: 28180992), which was not suitable for our analysis of mistrafficking. Therefore, we employed the weaker *Dorothy-GAL4* driver in combination with short-term induction *via* GAL80^{ts} to silence *Rab5*. In contrast, for silencing *sns*, the *Drosophila* nephrin, (Figure 5) our goal was to maximize silencing to assess whether the fishnet-like pattern depends on the presence of Sns. Residual

Sns protein may have caused partial persistence of the fishnet-like pattern. Thus, we used the stronger Prospero-GAL4 driver without GAL80 modulation for this analysis.

Previous studies (e.g., PMID: 18971929) showed that Sns depletion caused loss of slit diaphragm. Thus, it is expected that the fishnet is absent in *pros>sns-RNAi* nephrocytes.

We agree that the absence of the fishnet-like structure upon Sns depletion is expected. Nevertheless, this experiment importantly demonstrates that the fishnet-like molecular pattern is immediately dependent on the presence of *Drosophila* nephrin, directly linking the observed molecular pattern to this specific protein. This finding further supports that the structure is a genuine feature of the slit diaphragm, effectively ruling out an artifact. As mentioned in our response to Reviewer #1 (Question 5), we have now rephrased our conclusion on the *sns* silencing experiment to communicate the link between the fishnet pattern and the presence of Sns more clearly:

“We conclude that, consistent with previous findings^{21,22}, the presence of Sns is essential for the formation of the nephrocyte SD and is directly linked to the architecture of the newly discovered fishnet pattern.”

Will it provide more information if the authors use (Dot-Gal4, Gal80ts) to silence *sns* at developmental stages or adult stages?

We thank the reviewer for this suggestion. We performed additional experiments studying the SD structure after silencing of *sns* for 29 hours. At this stage, the levels of the *Drosophila* nephrin have been reduced to an intermediate level (**Figure 4, Supplementary Figure 5**). Partially removing the *Drosophila* nephrin resulted in shorter slit diaphragm lines revealing a fishnet-like pattern identical to the pattern in the wild-type control, demonstrating that the fishnet-like pattern forms while strictly maintaining a precise stoichiometry between nephrin and Neph1 orthologs in the molecular structure.

3. The images in Figure 4 panel J and supplementary Figure 2 panel B show high similarities. A clarification is missing in supplementary Figure 2 legend to show the genotypes. We apologize for not providing this information, the image has now been adapted and the genetic background has been included in the figure title: “**Supplementary Figure 3: Comparison of the resolution in nephrocytes between conventional room temperature transmission electron microscopy (RT-TEM) and cryo-electron tomography (cryo-ET) in wild-type animals.**”

4. It is challenging to appreciate the structure in figure 3D. For a comparison, the fishnet pattern proposed by Birtasu et al. 2023 is easy to understand. We thank the reviewer for pointing this out. Similar to Birtasu et al., we now provide three dimensional illustrations for our proposed models to facilitate understanding the structure in **Figure 3d**.

Minor comments:

1. In figure 1, panel F shows a region where two nephrocytes form contacts. This is different from the enlarged insets in panels E-E’. What is the rationale to choose the distinct regions? We have provided now many examples in the **Supplementary Figure 2. Figure 1f-g** (now **Figure 1h-i**) shows one of our best resolved tomograms. We hope that **Supplementary**

Figure 2, which displays raw data and different views will provide an improved overview. In addition we labeled **Figure 1f-g(h-i)** to specifically show two Nephrocytes.

2. In figure 1 F and G, the image contract does not seem to be optimal. Is there a way to improve it so that the reader can better appreciate the fine structures?

We thank the reviewer for calling our attention to that. We have now adjusted the contrast levels to improve visibility of the finer structures.

We would like to thank the reviewers! We have revised the transitions and explanations throughout the text to enhance readability.

Response to Reviewer #1:

Reviewer comment:

1) The light yellow coloring in Fig. 1H is not really visible.

Response:

We thank the reviewer for pointing that out. We have now adjusted it to make it more clearly visible.

Reviewer comment:

2) "Persistently" is spelled incorrectly on line 228.

Response:

Thank you for noting that, we have now corrected the spelling.

Reviewer comment:

*3) The transition between the section on partial silencing and more complete silencing of *sns* is confusing because it sounds like you are presenting something totally different rather than just doing the silencing longer. I think it needs to be just clearly stated in the first sentence that the previous silencing system was prolonged to get a more pronounced effect.*

Response:

Thank you for pointing this out; we adapted the manuscript text to distinguish more clearly between full and partial silencing:

The fishnet pattern is abolished upon prolonged *sns* silencing

To examine whether the fishnet pattern requires the presence of the SD protein *Sns*, we examined animals showing full, rather than partial, loss-of-function of *sns*. Without GAL80-dependent temporal restriction, the continuous, and therefore more pronounced, silencing of *sns* abolished the characteristic *Sns*-derived signal in light microscopy (Supplementary Figure 5). This confirms efficient silencing. When Kirre

lacked its binding partner entirely, its linear arrangement was replaced by a pattern of clusters (Figure 5a), while the mislocalized Kirre largely adhered to the surface (Figure 5b). The altered configuration confirms that the Neph1 ortholog alone is insufficient for SD formation and consequently, the characteristic labyrinthine channels were no longer formed (Figure 5c-e). In RT-TEM and cryo-ET acquisitions following such stronger, sustained silencing of *sns*, we observed electron-dense clusters decorating the cell membrane (Figure 5c,e,f), but the SDs with their fishnet pattern were abrogated. We conclude that, consistent with previous findings^{21,22}, the presence of *Sns* is essential for the formation of the nephrocyte SD and is directly linked to the architecture of the fishnet pattern shown here.

Reviewer comment:

4) *The way you discuss the ectopic formation of SD in labyrinthine channels is confusing to me. In the non-silenced sample, you described the correctly localized SD as being in labyrinthine channels so it is confusing to say that it then gets ectopically localized to labyrinthine channels.*

Response:

We thank the reviewer for this comment. Slit diaphragms form at the apical boundary of labyrinthine channels in wild-type animals near the cell surface. Thus, slit diaphragms are not being formed within the channel network per se in wild-type animals, which differs from ectopic slit diaphragms situated much deeper within the channels after *Rab5* silencing. We rephrased the manuscript to emphasize the distinction more clearly:

Visualization of the *Drosophila* filtration barrier *in situ*

The *Drosophila* nephrocyte shapes membrane invaginations into an elaborate network of labyrinthine channels. Entry into these channels is regulated by a filtration barrier comprising a basement membrane and a SD^{17,21,22}. The structural framework of the SD is established by *Sns* and *Kirre*, analogous to the function of their human orthologs nephrin and Neph1^{21,22} (Figure 1a). The SD, forming the apical boundary along the furrow-like labyrinthine channels, creates a fingerprint-like pattern in tangential sections (Figure 1b, Supplementary Figure 1a-b), which corresponds to a regular, dot-like appearance in cross-sections (Figure 1c, Supplementary Figure 1c; 3D visualization in Figure 1d, Supplementary Movie 1). The SD proteins *Sns* and *Kirre* colocalize at the entry points of the labyrinthine channels along the cell surface (Figure 1e-g, Supplementary Figure 1d-f).

Response to Reviewer #2:

We thank the reviewer for their positive feedback.